# GENERATIVE CLASSIFIERS
# AVOID SHORTCUT SOLUTIONS

**Alexander C. Li**
Carnegie Mellon University
alexanderli@cmu.edu

**Ananya Kumar**
Stanford University
ananya1@stanford.edu

**Deepak Pathak**
Carnegie Mellon University
dpathak@cs.cmu.edu

## ABSTRACT

Discriminative approaches to classification often learn shortcuts that hold in-distribution but fail even under minor distribution shift. This failure mode stems from an overreliance on features that are spuriously correlated with the label. We show that generative classifiers, which use class-conditional generative models, can avoid this issue by modeling all features, both core and spurious, instead of mainly spurious ones. These generative classifiers are simple to train, avoiding the need for specialized augmentations, strong regularization, extra hyperparameters, or knowledge of the specific spurious correlations to avoid. We find that diffusion-based and autoregressive generative classifiers achieve state-of-the-art performance on five standard image and text distribution shift benchmarks and reduce the impact of spurious correlations in realistic applications, such as medical or satellite datasets. Finally, we carefully analyze a Gaussian toy setting to understand the inductive biases of generative classifiers, as well as the data properties that determine when generative classifiers outperform discriminative ones.

## 1 INTRODUCTION

Ever since AlexNet (Krizhevsky et al., 2012), classification with neural networks has mainly been tackled with discriminative methods, which train models to learn $p_\theta(y \mid x)$. This approach has scaled well for in-distribution performance (He et al., 2016; Dosovitskiy et al., 2020), but these methods are susceptible to shortcut learning (Geirhos et al., 2020), where they output solutions that work well on the training distribution but may not hold even under minor distribution shift. The brittleness of these models has been well-documented (Recht et al., 2019; Taori et al., 2020), but beyond scaling up the diversity of the training data (Radford et al., 2021) so that everything becomes in-distribution, no approaches so far have made significant progress in addressing this problem.

In this paper, we examine whether this issue can be solved with an alternative approach, called generative classifiers (Ng & Jordan, 2001; Yuille & Kersten, 2006; Zheng et al., 2023). This method trains a class-conditional generative model to learn $p_\theta(x \mid y)$, and it uses Bayes' rule at inference time to compute $p_\theta(y \mid x)$ for classification. We hypothesize that generative classifiers may be better at avoiding shortcut solutions because their objective forces them to model the input $x$ in its entirety. This means that they cannot just learn spurious correlations the way that discriminative models tend to do; they must eventually model the core features as well. Furthermore, we hypothesize that generative classifiers may have an inductive bias towards using features that are *consistently predictive*, *i.e.*, features that agree with the true label as often as possible. These are exactly the core features that models should learn in order to do well under distribution shift.

Generative classifiers date back at least as far back as Fischer discriminant analysis (Fisher, 1936). Generative classifiers like Naive Bayes had well-documented learning advantages (Ng & Jordan, 2001) but were ultimately limited by the lack of good generative modeling techniques at the time. Today, however, we have extremely powerful generative models (Rombach et al., 2022; Brown et al., 2020), and some work is beginning to revisit generative classifiers with these new models (Li et al., 2023; Clark & Jaini, 2023). Li et al. (2023) in particular find that ImageNet-trained diffusion models exhibit the first "effective robustness" (Taori et al., 2020) without using extra data, which suggests that generative classifiers are have fundamentally different (and perhaps better) inductive biases. However, their analysis is limited to ImageNet distribution shifts and does not provide any understanding. Our paper focuses on carefully comparing deep generative classifiers against today's

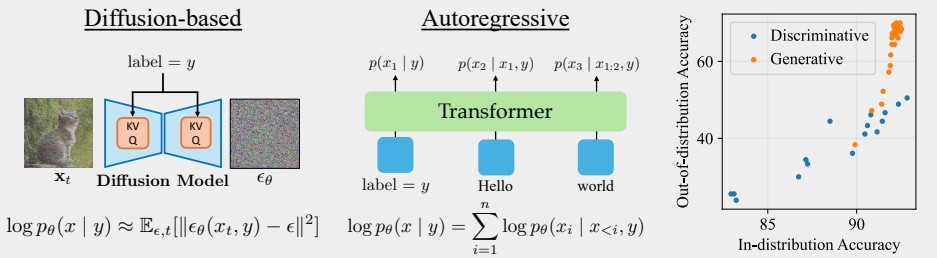

Figure 1: **Generative classifiers**. We repurpose today's best generative modeling algorithms for classification. Generative classifiers predict $\arg\max_y p_\theta(x \mid y)p(y)$. We use diffusion-based generative classifiers on image tasks and autoregressive generative classifiers on text tasks, and find that they scale better out-of-distribution than discriminative approaches.

discriminative methods on a comprehensive set of distribution shift benchmarks. We additionally conduct a thorough analysis of the reasons and settings where they work. We list our contributions:

- **Show significant advantages of generative classifiers on realistic distribution shifts**. Generative classifiers are simple and effective compared to previous distribution shift mitigations. They utilize existing generative modeling pipelines, avoid additional hyperparameters or training stages, and do not require knowledge of the spurious correlations to avoid. We run experiments on standard distribution shift benchmarks across image and text domains and find that generative classifiers consistently do better under distribution shift than discriminative approaches. Most notably, they are the *first algorithmic approach* to demonstrate "effective robustness" (Taori et al., 2020), where they do better out-of-distribution than expected based on their in-distribution performance (see Figure 1, right). We also surprisingly find better in-distribution accuracy on most datasets, indicating that generative classifiers are also less susceptible to overfitting.

- **Understand why generative classifiers work**. We carefully test several hypotheses for why generative classifiers do better. We conclude that the generative objective $p(x \mid y)$ provides more consistent learning signal by forcing the model to learn all features of $x$.

- **Provide insights from Gaussian data**. We compare generative (linear discriminant analysis) and discriminative (logistic) classification methods on a simplified setting. We find the existence of "generalization phases" that show which kind of approach does better, depending on the strength of the spurious correlations and noisy features in the data. These phases shed light on the inductive bias of generative classifiers towards low-variance features.

## 2 RELATED WORK

**Learning in the presence of spurious features**  It is been well-known that deep networks trained with empirical risk minimization (ERM) have a tendency to rely on spurious correlations to predict the label, such as the background in an image or the presence of certain words (Beery et al., 2018; Ribeiro et al., 2016; Geirhos et al., 2020; McCoy et al., 2019). Notably, overfitting to these shortcuts causes a degradation in performance under distribution shift, since these spurious correlations may no longer be predictive (Hendrycks & Dietterich, 2019; Rosenfeld et al., 2018; Taori et al., 2020). The performance on rare ("minority") groups in particular tends to suffer (Dixon et al., 2018; Zhao et al., 2017; Sagawa et al., 2019), and this imbalance is aggravated in highly overparametrized models (Sagawa et al., 2020). Theoretical works attribute this problem to the inductive bias of classifiers trained with cross-entropy loss; these classifiers prefer to find max-margin solutions, and thus fit spurious features even when they are not fully predictive like the core feature (Nagarajan et al., 2020; Puli et al., 2023). To address these failures in discriminative models, people use objectives that try to balance learning across different groups (Sagawa et al., 2019; Setlur et al., 2023; Lee et al., 2023), or add data augmentation to smooth out the spurious feature (Shen et al., 2022). However, these methods still tend to fail to capture the core feature and often lead to degradations in in-distribution performance. Some approaches focus on identifying the specific spurious features, annotating which examples contain them, and using that to rebalance the data (Wu et al., 2023; Ghosh et al., 2023; Kirichenko et al., 2022). Unfortunately, these approaches require significant manual effort, are not as scalable, and may not work for problems where humans do not understand the learned features.

Ideally, we find an approach with the right inductive bias to generalize well under distribution shift without requiring extra supervision.

**Classification with Generative Models** Few deep learning approaches have trained class-conditional generative models and used them directly for classification, perhaps due to the difficult task of modeling $p(x \mid y)$ with weaker generative models. However, recent generative models have significantly improved, especially with better techniques in diffusion probabilistic models (Sohl-Dickstein et al., 2015; Ho et al., 2020), and deep generative classification methods have recently been proposed (Li et al., 2023; Clark & Jaini, 2023). Li et al. (2023) showed that ImageNet-trained class-conditional diffusion models are competitive with discriminative classifiers and achieve the first nontrivial "effective robustness" (Taori et al., 2020) on ImageNet-A (Hendrycks et al., 2021) without using extra data. Prabhudesai et al. (2023) show that a hybrid generative-discriminative classifier can use test-time adaptation to improve performance on several synthetic corruptions. Other work (Clark & Jaini, 2023; Jaini et al., 2023) has shown that large pretrained generative models are more biased towards shape features and more robust to *synthetic* corruptions, but this may be due to effect of pretraining on extra data, or the fact that diffusion specifically confers resilience to input perturbations. Other works have found that generative classifiers can improve adversarial robustness (Grathwohl et al., 2020; Zimmermann et al., 2021; Chen et al., 2023; 2024a). However, *adversarial robustness* has been shown to not translate to *robustness to distribution shift* (Santurkar et al., 2020; Taori et al., 2020). Overall, it still remains unclear whether generative classifiers are more robust to the spurious correlations seen in *realistic* distribution shifts or *why* they might be better.

## 3 PRELIMINARIES

### 3.1 TYPES OF DISTRIBUTION SHIFT

We consider classification under two types of distribution shift. In subpopulation shift, there are high-level spurious features that are correlated with the label. For example, on CelebA (Liu et al., 2015), where the task is to predict whether a person's hair is blond or not blond, the spuriously correlated feature is the gender. This occurs because there are very few blond men in the dataset, so models typically learn to use the "man" feature. The spurious feature determines groups: the majority group contains examples where the spurious feature is correct, and the minority group contains examples where the spurious feature is incorrect. We also consider domain shift, where the test domain's data distribution is similar to the training domain's distribution. For example, training images in Camelyon17-WILDS (Koh et al., 2021) come from 3 hospitals, whereas the test images come from a disjoint 4th hospital. Spurious features that worked on the training distribution, *e.g.*, artifacts of the way slide staining or sample collection was done, may hurt accuracy under distribution shift (Veta et al., 2016; Komura & Ishikawa, 2018; Tellez et al., 2019). We examine 5 common distribution shift benchmarks in total: besides CelebA and Camelyon, we use Waterbirds ( Sagawa et al. (2019); subpopulation shift), FMoW (Koh et al. (2021); both subpopulation and domain shift), and CivilComments (Koh et al. (2021); subpopulation shift).

### 3.2 SHORTCOMINGS OF DISCRIMINATIVE CLASSIFIERS

Discriminative classifiers, which seek to maximize $p_\theta(y \mid x)$, can overly rely on the spurious features and fall victim to shortcut solutions (Geirhos et al., 2020). This is because they can use the spuriously correlated features to correctly and confidently fit the majority group examples. After this happens, the loss on these examples flattens out, and there is less gradient signal available to encourage the model to use core features (Li et al., 2019; Pezeshki et al., 2021). The model then overfits to the remaining minority examples where the spurious correlation does not help (Sagawa et al., 2020; Nagarajan et al., 2020). These shortcut solutions often work in-distribution but can fail, sometimes catastrophically, under even minor distribution shift. Significant effort has been put into preventing this, mainly by rebalancing the data so that the spurious correlation no longer holds (Sagawa et al., 2019; Kirichenko et al., 2022; Liu et al., 2021; Setlur et al., 2023). However, these methods all add additional hyperparameters and complexity to the training process, and often require knowledge of the exact distribution shift to counteract, which is impractical for realistic problems where there may be many spurious correlations.

## 4 GENERATIVE CLASSIFIERS

We now present generative classifiers, a simple paradigm for classification with class-conditional generative models. To classify an input $x$, generative classifiers first compute $p_\theta(x|y)$ with a class-conditional generative model and then utilize Bayes' theorem to obtain $p_\theta(y|x)$. This paradigm had been popular in machine learning with methods like linear discriminant analysis and Naive Bayes (Ng & Jordan, 2001), but has fallen out of favor in the modern era of deep learning. We revisit this paradigm with deep learning architectures and show its advantages for robustness to distribution shift in Section 5. Algorithm 1 gives an overview of the generative classification procedure.

### 4.1 INTUITION

Why could generative classifiers do better on these distribution shifts? In contrast to discriminative classifiers, which can minimize their training objective using just a few spurious features, generative classifiers need to model the entire input $x$. This means that they cannot stop at just the spurious features; their training objective requires them to learn both core and spurious features. This should translate to better training signal throughout the course of the training. We confirm this in Section 5.3. Note that learning both types of features does not mean that it uses them equally when classifying an input. The generative classifier should learn which type of features are more consistently correlated with the label and weight them accordingly. Section 6.3 and 6.4 demonstrates this inductive bias in a simple setting with Gaussian data.

### 4.2 DIFFUSION-BASED GENERATIVE CLASSIFIER

For image classification, we use diffusion models (Sohl-Dickstein et al., 2015; Ho et al., 2020), which are currently the state-of-the-art approach for image modeling. Diffusion models are trained to iteratively denoise an image and do not have an exact likelihood that can be computed in a single forward pass. They are typically trained with a reweighted variational lower bound of $\log p_\theta(x|y)$. To use them in a generative classification framework, we use that value to approximate $\log p_\theta(x \mid y)$:

$$\log p_\theta(x \mid y) \approx \mathbb{E}_{\epsilon,t}[\|\epsilon_\theta(x_t, y) - \epsilon\|^2] \tag{1}$$

Training the class-conditional diffusion models is done as normal – we use off-the-shelf training pipelines to train diffusion models from scratch, without modifying any hyperparameters. At inference time, we follow the Diffusion Classifier algorithm from (Li et al., 2023), which samples multiple noises $\epsilon$, adds them to the image to obtain noised $x_t = \sqrt{\bar{\alpha}_t}x + \sqrt{1 - \bar{\alpha}_t}\epsilon$, and does multiple forward passes through the network to obtain a Monte Carlo estimate of Eq. 1. This is done for each class, and the class with the highest conditional likelihood $\log p_\theta(x \mid y)$, which corresponds to the lowest denoising error, is chosen.

### 4.3 AUTOREGRESSIVE GENERATIVE CLASSIFIER

For text classification, we introduce generative classifiers built on autoregressive Transformer models, as they are the dominant architecture for text modeling. Since we need to now learn $p_\theta(x \mid y)$, where $x$ is a sequence of text tokens and $y$ is a label, we make a small modification to the training procedure. Instead of starting each sequence of text tokens with a "beginning of sequence" (BOS) token, we allocate $C$ special class tokens in our vocabulary, one per class, and replace BOS with the desired class token. Obtaining $\log p_\theta(x \mid y)$ can be done in a single forward pass:

$$\log p_\theta(x \mid y) = \log \left( \prod_{i=1}^{n} p_\theta(x_i \mid x_{<i}, y) \right) = \sum_{i=1}^{n} \log p_\theta(x_i \mid x_{<i}, y) \tag{2}$$

We train our Transformer as usual using cross-entropy loss over the entire sequence, with the ground truth label $y^*$ at the beginning. To classify a text sequence at inference time, we do $C$ forward passes, one with each possible class token. We then choose the class token with the lowest cross-entropy loss over the entire sequence as our prediction. Figure 1 (middle) shows a diagram of this method.

Overall, generative classifiers can be easily trained using existing generative modeling pipelines and do not require any specialized architectures, extra hyperparameters, data augmentation, multi-stage training, or knowledge of the specific shortcuts to avoid. Training is also cheap: all of our generative models can be trained on a single GPU in 2-3 days.

| Method | Waterbirds | | CelebA | | Camelyon | | FMoW | | CivilComments | |
|---|---|---|---|---|---|---|---|---|---|---|
| | ID | WG | ID | WG | ID | OOD | ID | OOD WG | ID | WG |
| ERM | 88.8 | 32.2 | 92.4 | 50.5 | 95.2 | 78.3 | 51.1 | 27.5 | **90.6** | 53.3 |
| LfF (Nam et al., 2020) | 86.4 | 28.9 | 90.8 | 34.0 | 90.5 | 66.3 | 49.6 | 31.0 | 87.9 | 49.4 |
| JTT (Liu et al., 2021) | 88.1 | 32.9 | 91.9 | 42.1 | 88.1 | 65.8 | 52.1 | 31.8 | 89.2 | 55.6 |
| RWY/DFR | 90.8 | 31.6 | **94.1** | 68.9 | 95.2 | 78.3 | 39.3 | 26.1 | 90.1 | 58.1 |
| Generative (ours) | **96.8** | **79.4** | 91.2 | **69.4** | **98.3** | **90.8** | **62.8** | **35.8** | 79.8 | **61.4** |

Table 1: **Accuracy on distribution shift benchmarks**. We show in-distribution (ID) and either worst-group (WG) or out-of-distribution (OOD) accuracy, depending on the type of shift in each dataset. Our generative approach performs the best on all five distribution shifts and 3/5 ID datasets.

## 5 EXPERIMENTS

We now compare our generative classification approach to discriminative methods that are commonly used today. We aim to answer the following questions in this section. First, do generative classifiers have better robustness to distribution shift? If so, why are they more robust than discriminative methods? We test multiple hypotheses to determine which explanation is correct.

### 5.1 SETUP

**Benchmarks** We use five standard benchmarks for distribution shift. Camelyon undergoes domain shift, so we report its OOD accuracy on the test data. Waterbirds, CelebA, and CivilComments undergo subpopulation shift, so we report worst group accuracy. FMoW has both subpopulation shift over regions and a domain shift across time, so we report OOD worst group accuracy. The first four are image benchmarks, while CivilComments is text classification. Waterbirds and CelebA are natural images, whereas Camelyon contains whole-slide images of cells and FMoW contains satellite images. In total, these benchmarks cover multiple shift types, modalities, and styles.

**Model Selection** We believe that it is unrealistic or impractical to know the exact distribution shift that will happen on the test set. Thus, we do not use knowledge of the spurious correlation or distribution shift when training or performing model selection, and instead tune hyperparameters and perform early stopping on the in-distribution validation accuracy, not the worst-group accuracy. This is the most valuable setting, as it matches how models are often deployed in practice, and thus is a popular experimental setting for evaluating methods (Koh et al., 2021; Yang et al., 2023; Liu et al., 2021; Setlur et al., 2023). We use class-balanced accuracy for model selection as it uniformly improves performance on each dataset for all methods (Idrissi et al., 2022).

**Baselines** We compare generative classifiers against several discriminative baselines. ERM minimizes the average cross-entropy loss of the training set and is the standard method for training classifiers. We additionally evaluate several methods designed to combat spurious features. Learning from Failure (LfF) (Nam et al., 2020) simultaneously trains one network to be biased and uses it to identify samples that a second network should focus on. Just Train Twice (JTT) (Liu et al., 2021) is a similar two-stage method that first trains a standard ERM model for several epochs, and then heuristically identifies worst-group examples as training points with high loss under the first model. JTT then upsamples these points and trains a second classifier. DFR (Kirichenko et al., 2022) fine-tunes a model on a training set that has been carefully balanced to make the spurious feature unpredictive. As in prior work (Yang et al., 2023), DFR samples data from each class equally when there are no spurious feature annotations. This is equivalent to RWY (Idrissi et al., 2022) and can help if there is class imbalance related to the spurious correlation. For fairness, we train all models, generative and discriminative, from scratch to eliminate the effect of differing pre-training datasets.

**Models** For image-based tasks, all discriminative baselines use ResNet-50, ResNet-101, and ResNet-152, whereas our generative classifier approach trains a class-conditional U-Net-based latent diffusion model (Rombach et al., 2022). For text-based tasks, all discriminative baselines use an encoder-only Transformer, whereas our generative classifier approach trains a Llama-style autoregressive language model (Touvron et al., 2023) from scratch. See Appendix B for more details.

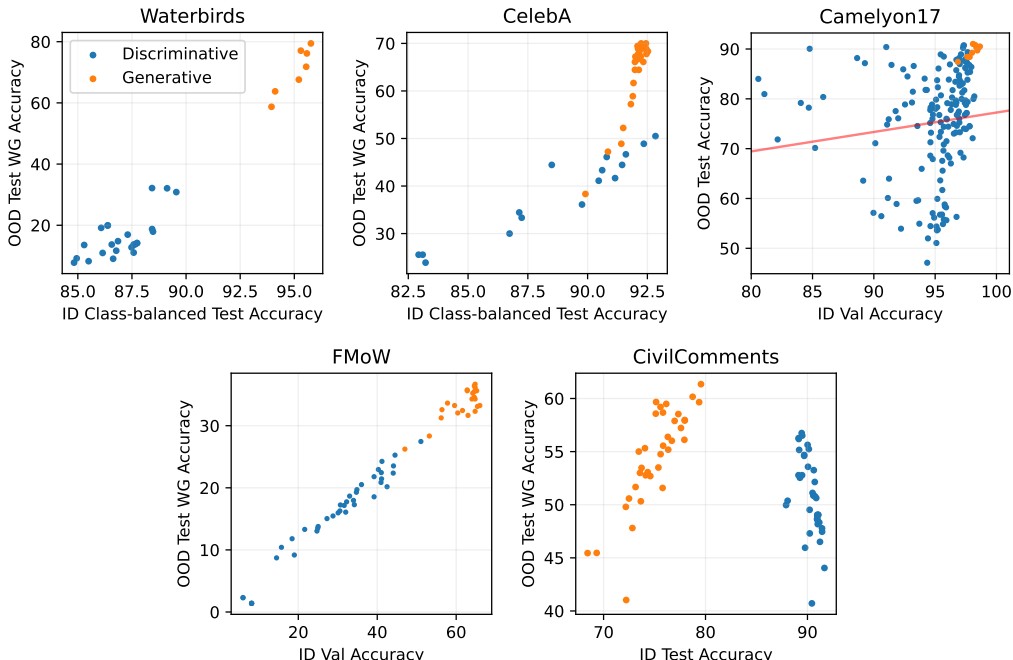

Figure 2: **In-distribution vs out-of-distribution accuracy** for each dataset. Each point corresponds to a separate training run, other than the diffusion-based generative classifier results, which are checkpoints of a run with default training hyperparameters. We observe better OOD scaling trends (i.e., effective robustness) for generative classifiers on CelebA, CivilComments, and potentially Camelyon17, although results are noisy for this dataset (the red line in Camelyon17 denotes a linear fit for the relationship between ID and OOD accuracy for discriminative models). On the remaining two datasets, they follow the same trend and do better both ID and OOD.

## 5.2 RESULTS ON DISTRIBUTION SHIFT BENCHMARKS

**Main Results**    Table 1 compares generative classifiers against discriminative baselines on the distribution shift benchmarks. Compared to the discriminative baselines, generative classifiers have better worst-group or OOD accuracy on all five datasets. Surprisingly, generative classifiers also achieve *significantly* better in-distribution accuracy on three of the five datasets, which indicates less overfitting. These results suggest that generative classifiers may have an advantage in both (a) learning core features that generalize across distribution shifts, and (b) learning features that generalize from the training set to the ID test set.

**Accuracy above the line**    Comparing the best generative classifier against the best discriminative classifier provides a one-dimensional understanding of each approach. To provide a better sense of which method may scale better in the future, Figure 2 plots the in-distribution and out-of-distribution accuracies of each family of methods. We can classify the benchmarks into two sets:

1. Generative classifiers are better both ID and OOD, and lay on the same trend line as discriminative models. This includes Waterbirds and FMoW.

2. Generative classifiers have a significantly better OOD performance *trend*, but are the same or worse in-distribution. This includes CelebA, CivilComments, and potentially Camelyon.

The second case, where generative classifiers have better OOD accuracy than discriminative classifiers at any ID accuracy, demonstrates "effective robustness" (Taori et al., 2020). This suggests fundamentally better out-of-distribution behavior for generative classifiers in some scenarios and indicates that they may be the right approach to classification after further scaling. Our results corroborate findings from Li et al. (2023), which found some signs of effective robustness on ImageNet. Section 6 examines a toy setting and provides insights into the cause of this "effective robustness."

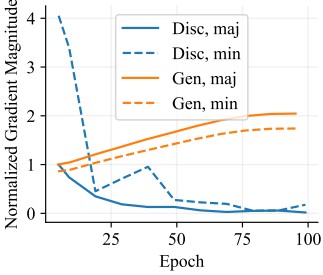

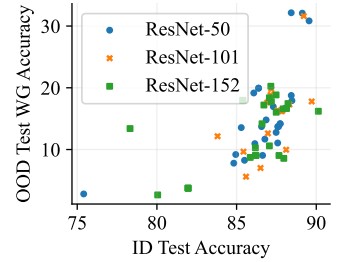

| Train Objective | ID | WG |
|---|---|---|
| $p(y \mid x)$ | 91.4 | 35.7 |
| $p(x)$ and $p(y \mid x)$ | **91.7** | 35.4 |
| $p(x \mid y)$ (ours) | 79.8 | **61.4** |

Table 2: **Alternative training objectives** for an autoregressive model on CivilComments. $p(y \mid x)$ is a standard discriminative approach with cross-entropy loss, and "$p(y \mid x)$ and $p(x)$" tests if adding an unconditional generative modeling improves performance.

Figure 3: **Gradient norms**. Gradient for disc. model rapidly decays to 0, so its learning signal is reduced, while it does not decay for generative classifier.

Figure 4: **Scaling disc. model size** does not improve accuracy on Waterbirds. This shows that model size is *not* a confounder in our experiments.

### 5.3 WHY DO GENERATIVE CLASSIFIERS DO BETTER?

We test several hypotheses for how generative classifiers outperform the discriminative baselines.

**Learning More from Majority Examples**    Our intuition is that the generative objective $\log p_\theta(x \mid y)$ provides more consistent learning signal across epochs. In contrast, discriminative models may use spurious features to make confident and correct predictions on the training set and lose the gradient signal necessary to use the core features. We test this by measuring the gradient norm on majority and minority examples across epochs. We compute the per-example gradient norm $\|\nabla_\theta \mathcal{L}(x_i, y_i)\|_2$ and average it over the majority and minority groups. We normalize this by the average majority group gradient norm at epoch 5 in order to fairly compare different architectures that have different loss landscapes. Figure 3 shows these metrics on CivilComments with toxic comments about the Black demographic as the minority group. For the discriminative model, the majority group gradient quickly vanishes, and the minority group gradient starts high but eventually decays. The generative classifier, however, has very similar gradient norm across the majority and minority groups, and the gradient norm actually slightly increases over training. These results support our intuition that the generative objective helps the model learn more from examples with and without the spurious features. Note: the *per-example* gradient norm for generative classifiers *does not* go to 0, since there is always a way to increase the likelihood of a single data point.

**Does an unconditional objective $p(x)$ improve discriminative performance?**    One hypothesis is that the generative classification objective $p_\theta(x \mid y)$ teaches the model better features in general, similar to how generative pre-training methods (Devlin et al., 2018; He et al., 2022) learn features that are useful for fine-tuning. We test this on CivilComments, as the architecture makes it simple to add an unconditional generative objective $p(x)$. Instead of placing the class-specific token at the beginning of the sequence, we place it at the end. Predicting the text tokens of $x$ now corresponds to predicting $p(x)$, and predicting the class-specific token at the end corresponds to $p(y \mid x)$. Table 2 shows that adding the objective $p(x)$ does not affect performance, so we reject this hypothesis.

**Model Size**    In our image classification experiments, our generative classifier used a standard 395M parameter UNet (Rombach et al., 2022), which is far more than the 26M parameters in its ResNet-50 (He et al., 2016) discriminative counterpart. Could the greater parameter count could be responsible for the difference in performance and OOD behavior? We first note that the discriminative classifier used for CivilComments in Table 1 contains 67M parameters, which is more than the 42M parameters we use in our autoregressive generative classifier. Furthermore, the architectures and parameter counts of the discriminative $p(y \mid x)$ and generative $p(x \mid y)$ classifiers are exactly matched in Table 2. On image tasks, we test whether parameter count matters by scaling from ResNet-50 up to ResNet-152 (He et al., 2016). We perform a sweep over model size and training hyperparameters (learning rate and weight decay). Figure 4 and Figure 8 show that increasing discriminative model size does not improve performance. This aligns with previous work: Sagawa et al. (2020) found that increasing discriminative model size can actually *hurt* OOD performance.

## 6 ILLUSTRATIVE SETTING

We explore a simplified Gaussian data setting and find that linear generative classifiers can also display better robustness to distribution shift compared to their discriminative counterparts. We rigorously explore this behavior in order to understand the inductive bias of generative classifiers. Finally, we connect our findings back to practice and explain the varying empirical behavior for generative vs discriminative classifiers.

### 6.1 DATA

Consider binary classification with label $y \in \{-1, +1\}$. The features are $x = (x_{\text{core}}, x_{\text{spu}}, x_{\text{noise}}) \in \mathbb{R}^d$, where:

$$x_{\text{core}} \mid y = \mathcal{N}(y, \sigma^2) \in \mathbb{R} \tag{3}$$

$$x_{\text{spu}} \mid y = y\mathcal{B} \text{ w.p. } \rho, \text{ else } -y\mathcal{B} \in \mathbb{R} \tag{4}$$

$$x_{\text{noise}} \mid y = \mathcal{N}(0, \sigma^2_{\text{noise}}) \in \mathbb{R}^{d-2} \tag{5}$$

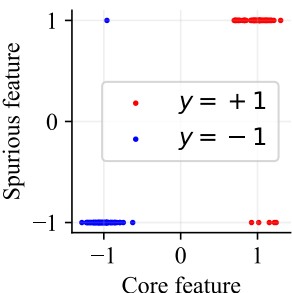

We set the spurious correlation ratio $\rho = 0.9$ and core feature standard deviation $\sigma = 0.15$, which is small enough that the data can be perfectly classified by using only the core feature $x_{\text{core}}$ and ignoring the remaining features. Figure 5 shows a visualization of the core and spurious features. The majority groups consist of samples where the spurious and core features agree (top right and bottom left of Fig. 5), and the minority groups consist of samples where the spurious and core features disagree (top left and bottom right).

Figure 5: Visualization of features (noise dims not shown).

This synthetic dataset has previously been used to understand the failure modes of discriminative classifiers in previous work (Sagawa et al., 2020; Idrissi et al., 2022; Setlur et al., 2023) and is a natural simplified setting for us to study the advantages of generative classifiers.

### 6.2 ALGORITHMS

**Discriminative**  We analyze unregularized logistic regression, as is done in previous work (Sagawa et al., 2020; Nagarajan et al., 2020). Since the data is linearly separable, logistic regression learns the max-margin solution when trained via gradient descent (Soudry et al., 2018).

**Generative**  We use linear discriminant analysis (LDA), a classic generative classification method that models each class as a multivariate Gaussian. It fits separate class means $\mu_{-1}$ and $\mu_{+1}$ but learns a shared covariance matrix $\Sigma$ for both classes. Assuming balanced classes, LDA predicts:

$$\arg\max_y p(x \mid y) = \text{sign}\left(\log \frac{p(x \mid y = +1)}{p(x \mid y = -1)}\right) = \text{sign}\left(\log \frac{\mathcal{N}(x \mid \mu_{+1}, \Sigma)}{\mathcal{N}(x \mid \mu_{-1}, \Sigma)}\right) \tag{6}$$

This corresponds to a linear decision boundary with coefficients $w_{LDA} = \Sigma^{-1}(\mu_{+1} - \mu_{-1})$. We chose LDA because it has the least inductive bias among common linear generative classifiers (*e.g.*, it is equivariant to rotations). For this reason, we rejected methods like Naive Bayes, which learns an axis-aligned generative model, even though theoretical analysis would have been easier.

### 6.3 THE INDUCTIVE BIAS OF LDA

We carefully examine a setting where the generative approach has the same in-distribution performance, but it outperforms the discriminative approach on the worst group (OOD). Figure 6 compares the behavior of LDA and logistic regression on toy data with data dimension $d = 1026$ and noise variance $\sigma^2_{noise} = 0.36$. We find that both methods have similar in-distribution accuracies, but LDA does significantly better on the minority group. In fact, Figure 6 (middle) shows that LDA has essentially no performance gap between the majority and minority groups, which indicates that it *does not use the spurious feature at all*. In contrast, logistic regression has a large performance gap between the groups. This can be explained by looking at the linear coefficients learned by both methods. Figure 6 (right) shows the ratio $|w_{spu}|/|w_{core}|$ between the weights on the shortcut and

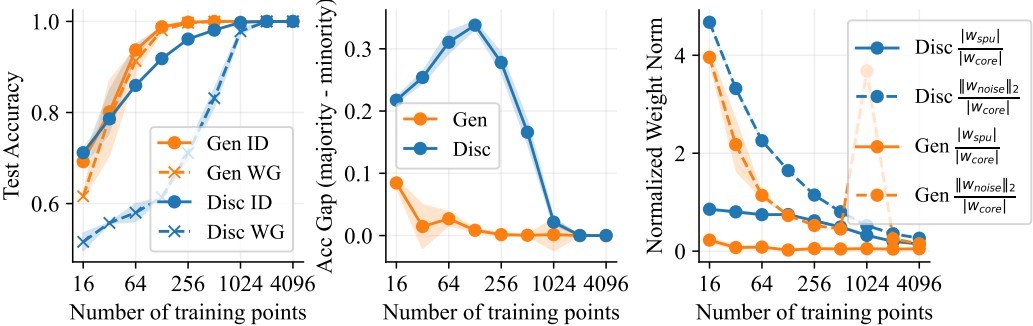

Figure 6: **Illustrative setting for shortcut learning. Left**: in-distribution accuracies are roughly the same between generative (LDA) and discriminative (logistic regression) methods, but LDA achieves much higher minority group accuracy. **Middle**: the difference between the majority and minority test accuracies as a function of the number of training examples. The generative method displays better robustness to the spurious correlation. **Right**: spurious feature weights $|w_{spu}|$ and noise feature weights $\|w_{noise}\|_2$, normalized by the magnitude of the core feature weight $w_{core}$. LDA puts much less weight on the spurious feature, even with very little training data. Logistic regression puts more weight on the noise, and only achieves good in-distribution accuracy by using the spurious feature. Shading denotes $\pm 1$ standard deviation over 25 seeds.

core features. Ideally, this ratio goes to 0 as fast as possible as the model sees more data. Logistic regression, however, places significant weight on the spurious feature until it gets thousands of training examples. LDA is far more data-efficient and places almost no weight on the spurious feature with as few as 16 training examples. Interestingly, logistic regression puts more weight on the noisy features than LDA does. It is only by putting significant weight on the spurious feature that it achieves good in-distribution performance, though this hurts worst group accuracy.

These differences in behavior indicate that LDA has a significantly different inductive bias. We suspect that the most important factor is the core feature variance $\sigma^2$. We increased $\sigma$ from 0.15 to 0.6 and reran the same analysis. Figure 14 shows that LDA now consistently underperforms logistic regression, both in-distribution and out-of-distribution. Why is this? Intuitively, when the learned probability $p_\theta(x_i \mid y^*)$ of feature $x_i$ is low (*i.e.*, the feature is not consistently correlated with the label) compared to other features, the generative classifier downweights this feature in its prediction. This helps improve robustness to distribution shift, since, by definition, we believe that the core feature $x_{core}$ should be the most consistently predictive.

## 6.4 GENERALIZATION PHASE DIAGRAMS

The feature dimension $d$, spurious feature scale $\mathcal{B}$, noisy feature variance $\sigma^2_{noise}$, and core feature variance $\sigma^2$ influence the solutions that models prefer to learn. Intuitively, the spurious feature scale $\mathcal{B}$ controls the saliency of the shortcut feature, and larger $\mathcal{B}$ makes it easier for the model to learn this shortcut. The noisy feature variance $\sigma^2_{noise}$ controls how easy it is for a model to overfit to training examples (Nagarajan et al., 2020). The core feature variance $\sigma^2$ controls how consistently the core feature predicts the label. Varying these properties of the data creates a family of datasets, and we use them to understand when generative classifiers outperform their discriminative counterparts.

Each plot in Figure 7 varies $\mathcal{B}$ and $\sigma^2_{noise}$, for a given number of training examples $n = 32$ and $d = 1024$. Each successive subplot corresponds to increasingly larger variance $\sigma^2$ in the core feature, and each plot is divided into regions depending on which method does better ID or OOD for the given $(\mathcal{B}, \sigma^2_{noise})$ at that location. We call this a *generalization phase diagram*, since it resembles a phase diagram which shows the impact of pressure and temperature on the physical state of a substance. In our case, there are four possible generalization phases:

1. The generative classifier is better both ID and OOD. This typically happens at high $\sigma^2_{noise}$, since the discriminative model overfits using the noise features.

2. The discriminative classifier is better both ID and OOD. This happens at low $\sigma^2_{noise}$.

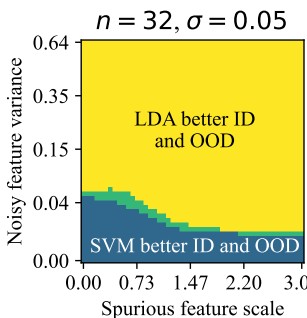 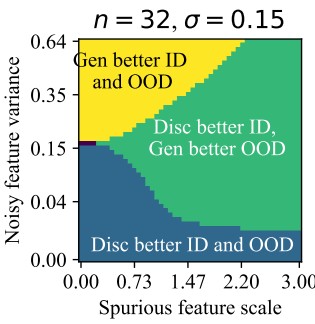 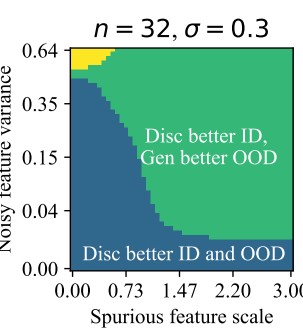

Figure 7: **Generalization phase diagrams**. We vary the scale $\mathcal{B}$ of the spurious feature and the variance $\sigma^2_{\text{noise}}$ of the noise features and evaluate their effect on the ID and OOD test accuracy of generative classifiers (LDA) vs discriminative classifiers (logistic regression). Each plot corresponds to a different variance $\sigma^2$ of the core feature, and the color of each pixel denotes which classifier does better for a particular combination of $\mathcal{B}$ and $\sigma^2_{\text{noise}}$. We observe three main phases of generalization: (1) discriminative has better ID and OOD accuracy, (2) generative has better ID and OOD accuracy, and (3) discriminative does better ID and generative does better OOD. Each plot corresponds to a different standard deviation $\sigma$ of the core feature. As $\sigma$ increases, the core feature becomes less reliable, and the generative classifier uses the spurious and noise features more. This shows the inductive bias of generative classifiers: they prefer *consistently predictive features*.

3. The discriminative classifier is better ID, but the generative classifier is better OOD. This phase occurs at a sweet spot of $\mathcal{B}$ and $\sigma^2_{noise}$. Moderate noise allows some overfitting, but the spurious feature is strong enough for the discriminative model to achieve good ID accuracy. However, its heavy reliance on the spurious feature reduces its OOD performance.

4. The generative classifier is better ID, but the discriminative classifier is better OOD. This is exceedingly rare (see the dark, unlabeled regions in Figure 7).

Notably, there is no free lunch. Even in this setting, neither generative nor discriminative classifiers are uniformly better than the other. However, we do note that $\mathcal{B}$ and $\sigma^2_{noise}$ are unbounded above, and generative classifiers should do comparatively better as the strength of shortcuts or noise increases.

Finally, while it is hard to map $\mathcal{B}$ and $\sigma^2_{noise}$ directly onto a realistic image or text dataset, they do offer insights on important properties of the data that determine which method is suitable for a given task. Indeed, we can categorize the distribution shift benchmarks into these phases based on their generative vs discriminative behavior. Waterbirds and FMoW fall in phase 1 (generative better ID and OOD), CelebA and CivilComments fall in phase 3 (discriminative better ID and generative better OOD), and Camelyon lies on the transition boundary between phase 1 and 3, since the generative classifier achieves better OOD and similar ID accuracy compared to discriminative baselines.

## 7 CONCLUSION

Discriminative approaches to classification have dominated the field since AlexNet catalyzed the widespread adoption of deep learning. Despite their prevalence, these methods face increasing limitations, including vulnerability to distribution shift and escalating data requirements. In this paper, we present a simple alternative. We revisit the paradigm of generative classifiers and show that they have significant advantages in both in-distribution and out-of-distribution performance on realistic distribution shift benchmarks. We carefully analyze their behavior, and finally show insights from a simplified, illustrative setting into when generative classifiers can be expected to do better.

As deep generative classifiers have not been well-explored, there is significant room for future work. The inference cost of these generative classifiers, especially diffusion-based ones, is currently impractically high. It is also unclear how to incorporate complex augmentations, such as Mixup, into generative classifiers. Finally, the ideas from this work may be useful in other contexts, such as language modeling. Tasks like sentiment analysis, code completion, and reasoning are currently being done in a discriminative approach: given a context $x$, predict the correct answer $y$ by sampling from $p_\theta(y \mid x)$. Improving the performance and out-of-distribution robustness of these models by doing a generative approach $p(x \mid y)$ would be a particularly exciting direction.

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

APPENDIX

# A    ADDITIONAL ANALYSIS

## A.1    ADDITIONAL RESULTS ON THE EFFECT OF DISCRIMINATIVE MODEL SIZE

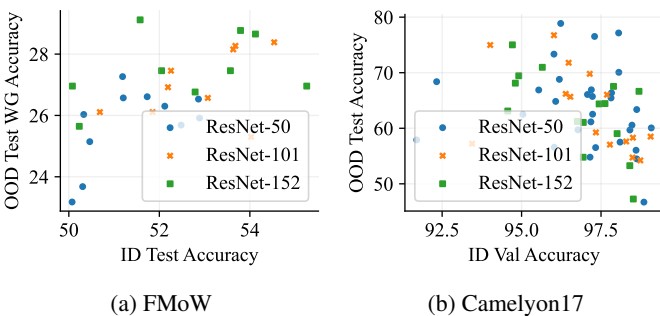

(a) FMoW                                    (b) Camelyon17

Figure 8: **Scaling up discriminative model size does not improve performance**. Each point with the same color is a model trained with different hyperparameters (learning rate and weight decay). Results on Waterbirds are shown in Figure 4.

We add additional results to our investigation into the role of discriminative model size. Previously, our analysis of CivilComments in Section 5.3 showed that matching the parameter count between the discriminative and generative classifiers did not account for the qualitative differences in their generalization behavior. Furthermore, Figure 4 showed that increasing model size on Waterbirds did not improve performance. Figure 8 shows additional results. On FMoW, scaling only helps when going from ResNet-50 to ResNet-101; further scaling did not help. On Camelyon17, increasing model size had no effect on performance. Overall, we can confidently conclude that model size is not responsible for generative classifiers' improved robustness to distribution shift.

## A.2    SCALING CAN IMPROVE GENERATIVE CLASSIFIERS

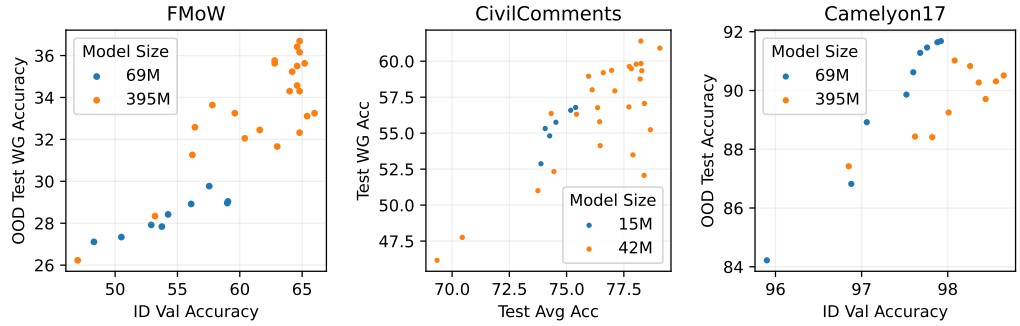

Figure 9: **Effect of scaling up generative classifiers**. Increasing the number of parameters helps significantly on FMoW and CivilComments, but can sometimes hurt: OOD accuracy drops on Camelyon17 with a larger generative classifier.

Scaling model size has proved extremely effective for generative models in other settings (Brown et al., 2020; Kaplan et al., 2020; Hoffmann et al., 2022). This has typically been done in the "almost infinite data" regime, where only a few epochs are used, and overfitting is not an issue. Does scaling similarly help here for our generative classifiers?

We tried different model scales on three of our distribution shift benchmarks: FMoW, CivilComments, and Camelyon17. Figure 9 shows the results of our investigation. On FMoW and CivilComments, scaling model size significantly improves performance both in- and out-of-distribution. However, on Camelyon17, a smaller model actually does significantly better out-of-distribution than a model that is 5.5 times as large. This indicates that overfitting can become an issue in this setting,

where we have limited training data and must be careful about overfitting. Overall, we are excited by the fact that scaling generative classifiers *can be beneficial* in some settings, unlike discrimiminative classifiers, which consistently show poor use of extra model capacity (see Figure 4, Table 2, Figure 8).

## A.3 RESULTS ON ADDITIONAL DATASETS

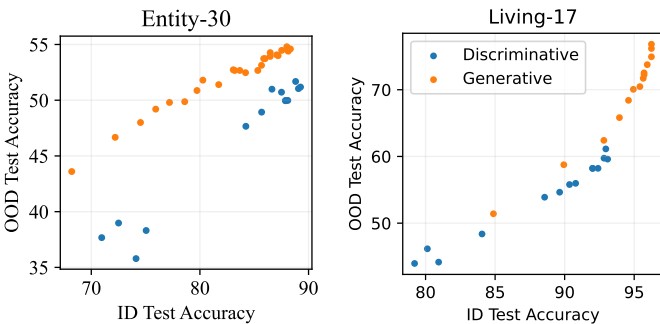

Figure 10: **In-distribution vs out-of-distribution accuracy** for additional subpopulation shift datasets (Santurkar et al., 2020). We again observe OOD scaling trends for generative classifiers. Each point for a discriminative model corresponds to a separate model with a different architecture, augmentation, or adversarial training method. Accuracies for the discriminative models are taken from Santurkar et al. (2020).

We additionally run experiments on two highly-used subpopulation shift benchmarks from BREEDS (Santurkar et al., 2020): Living-17 (with 17 animal classes) and Entity-30 (with 30 classes). As usual, we train our diffusion-based generative classifiers from scratch on each training set and evaluate them on the in-distribution and out-of-distribution test sets. We compare against discriminative baselines reported in the original BREEDS paper, which includes interventions such as stronger augmentations or adversarial training. Figure 10 displays the same trends here as our main results. Both datasets display effective robustness (for a given ID accuracy, the OOD accuracy of the generative classifier is higher), though the effect is much stronger on Entity-30.

## A.4 CORRELATION BETWEEN GENERATIVE AND DISCRIMINATIVE PERFORMANCE

We take a careful look at how well generative capabilities like validation likelihood and sample quality correlate with classification performance. Figure 11 shows how these three metrics evolve over the course of training for a diffusion-based generative classifier on CelebA.

We first find that the model does not need to generate good samples in order to have high classification accuracy. The first generation in Figure 11 has significant visual artifacts, yet the generative classifier already achieves 90% class-balanced accuracy. This makes sense: for ground-truth class $y^*$, the classifier only needs $p_\theta(x \mid y^*) > p_\theta(x \mid y)$ for all other classes $y \neq y^*$, so $p_\theta(x \mid y^*)$ can be low as long as $p_\theta(x \mid y \neq y^*)$ is even lower. In fact, given a generative classifier $p_\theta(x \mid y)$, one can construct another generative classifier $\tilde{p}(x \mid y) = \lambda p_\theta(x \mid y) + (1 - \lambda)p_{\text{other}}(x)$, which has the same accuracy as $p_\theta$ but generates samples that look increasingly like $p_{\text{other}}$ as $\lambda \to 0^+$.

However, even though sample quality is not necessary for high accuracy, we do find that validation diffusion loss correlates well with class-balanced accuracy. As the loss decreases, class-balanced accuracy correspondingly increases. Figure 12 shows how an increase in validation diffusion loss due to overfitting translates to a corresponding decrease in classification accuracy on Waterbirds.

Finally, Figure 11 shows how we can check the samples to audit how the generative classifier models the spurious vs core features. The samples are generated deterministically with DDIM (Song et al., 2020) from a fixed starting noise, so the sample from the last checkpoint shows that the model is increasing the probability of blond men (the minority group in CelebA). This means that the model is modeling less correlation between the hair color (causal for the blond vs not blond label) and the gender (the shortcut feature). This is one additional advantage of generative classifiers: generating samples is a built-in interpretability method (Li et al., 2023). Again, as we note above, generation of a specific feature is sufficient but not necessary to show that it is being used for classification.

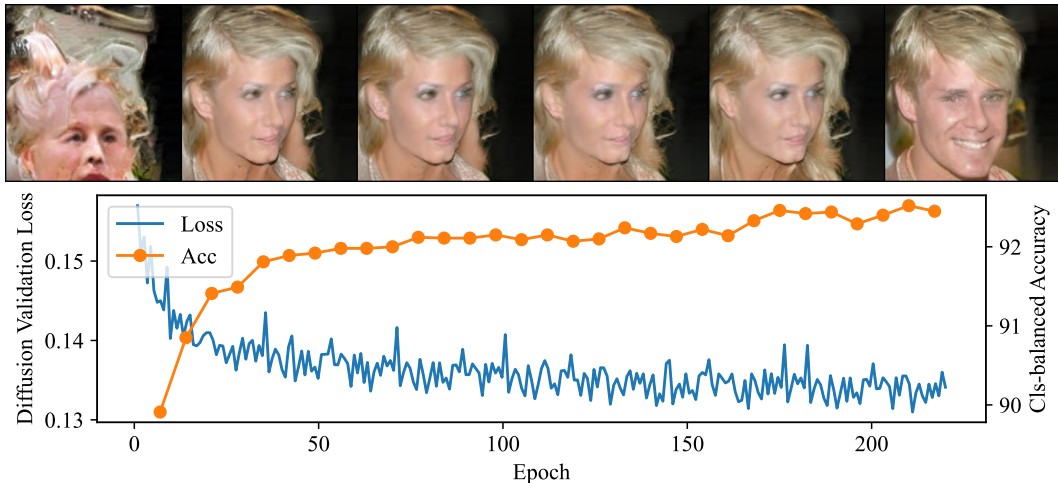

Figure 11: **Correlation between accuracy and generative performance. Top**: class-conditional DDIM samples generated from the same noise using intermediate checkpoints. **Bottom**: diffusion validation loss and class-balanced accuracy on CelebA by training epoch. **Main findings**: First, high classification accuracy can be achieved even without good sample quality (see the first generation). Second, generative validation loss is highly correlated with classification accuracy. Third, as training progresses, the minority group (blond men) becomes more likely, indicating that the generative classifier correctly models less correlation between hair color (causal) and gender (shortcut).

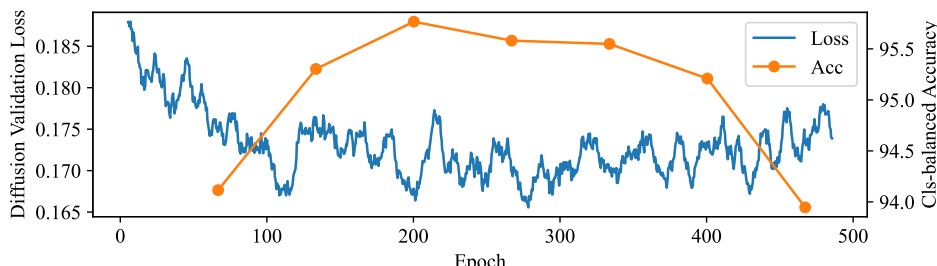

Figure 12: Overfitting in diffusion loss on Waterbirds directly translates to overfitting in classification accuracy. We smooth the loss for better visual clarity.

## A.5 EFFECT OF IMAGE EMBEDDING MODEL

For our image results in the main paper, we trained latent diffusion models from scratch for each dataset. In order to be consistent with the generative modeling literature and keep the diffusion model training pipeline *completely unmodified*, we trained the diffusion models on the latent space of a pre-trained VAE (Rombach et al., 2022). This VAE compresses $256 \times 256 \times 3$ images into $32 \times 32 \times$

| Embedding model | Waterbirds | | CelebA | | Camelyon | |
|---|---|---|---|---|---|---|
| | ID | WG | ID | WG | ID | OOD |
| Pre-trained VAE (Rombach et al., 2022) | **96.8** | **79.4** | 91.2 | 69.4 | 98.3 | 90.8 |
| PCA patch embeddings (Chen et al., 2024b) | 93.8 | 61.7 | **91.3** | **71.1** | **98.7** | **93.8** |

Table 3: **Effect of image embedding model.** We compare different image encoders, which map the image from $256 \times 256 \times 3$ to $32 \times 32 \times 4$. For our main results, we use the pre-trained deep VAE released in the original LDM paper (Rombach et al., 2022). We compare it to a PCA-based patch embedding that tokenizes each $8 \times 8 \times 3$ patch independently and is trained separately on each dataset. We find that the pre-trained VAE is not consistently better, as it only does better on 1 of the 3 datasets that we tested the PCA encoder on.

4 latent embeddings, which are cheaper to model. Perhaps our generative classifier is benefiting from an encoder that makes use of extra pre-training data? We test this hypothesis by trying to remove as much of the pre-trained encoding as possible. Following previous analysis work on diffusion models (Chen et al., 2024b), we replace the VAE network with a simple PCA-based encoding of each image patch. Specifically, we turn each image into $32 \times 32$ total $8 \times 8 \times 3$ pixel patches, and use PCA to find the top 4 principal components of the patches. When encoding, we normalize by the corresponding singular values to ensure that the PCA embeddings have approximately the same variance in each dimension. Overall, we perform this process separately on each training dataset, which completely removes the effect of pre-training, and train a diffusion model for each dataset within the PCA latent space. Table 3 compares this embedding model to the VAE and finds that it actually performs *better* on 2 of the 3 datasets. We conclude that the pre-trained encoder does not have a significant directional effect on our generative classifier results.

## A.6 COMPARISON WITH PRE-TRAINED DISCRIMINATIVE MODELS

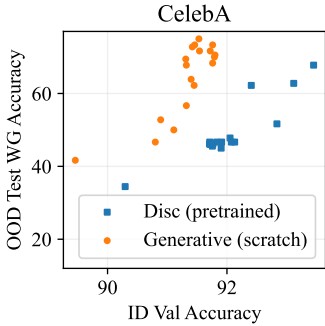

Figure 13: Finetuning a pretrained discriminative model improves performance, but it still does not achieve the same "effective robustness" as our generative classifier.

All of our experiments so far train the classifier (whether discriminative or generative) from scratch. This is done to ensure a fair, apples-to-apples comparison between methods. What happens if we use a pretrained discriminative model? In preliminary comparisons, we use a ResNet-50 pretrained with supervised learning on ImageNet-1k (Krizhevsky et al., 2012) and finetune it on CelebA. Figure 13 shows the results of this *unfair comparison* between a pretrained discriminative model versus our generative classifier trained from scratch. We find that pretraining helps, but it does not significantly close the gap with the generative classifier. This is in spite of the fact that the discriminative model has seen an extra 1.2 million labeled training images, those labels have more bits (since there are 1000 classes instead of just two), and the pretraining classification task has minimal spurious correlations that are relevant to the downstream task.

## A.7 ADDITIONAL PLOTS FOR GENERALIZATION PHASE DIAGRAMS

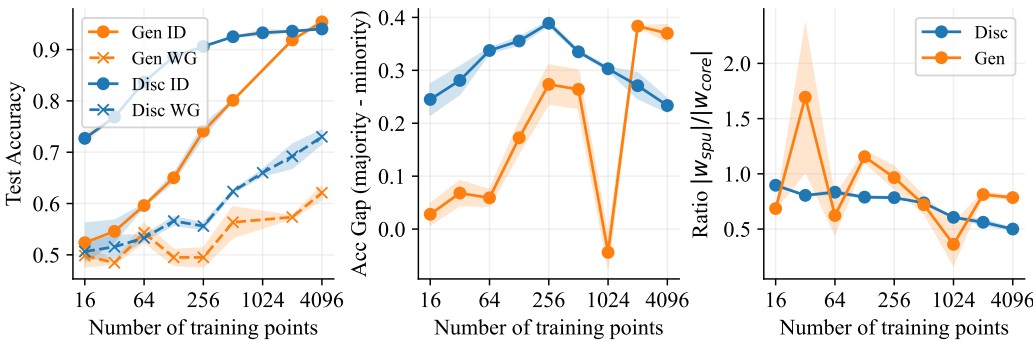

Figure 14: Comparing logistic regression and LDA when the core feature variance has been increased from $\sigma = 0.15$ to $\sigma = 0.6$. The generative approach's accuracy drops much more in this setting.

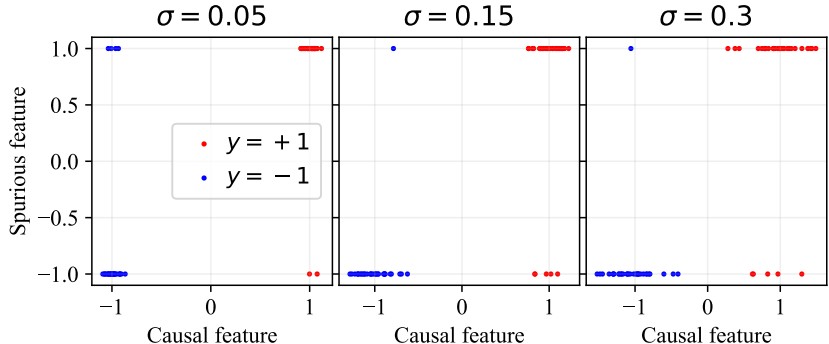

Figure 15: Effect of varying the standard deviation $\sigma$ of the core feature. $d - 2$ noise dimensions not shown. These correspond to the $\sigma$ shown in Figure 7.

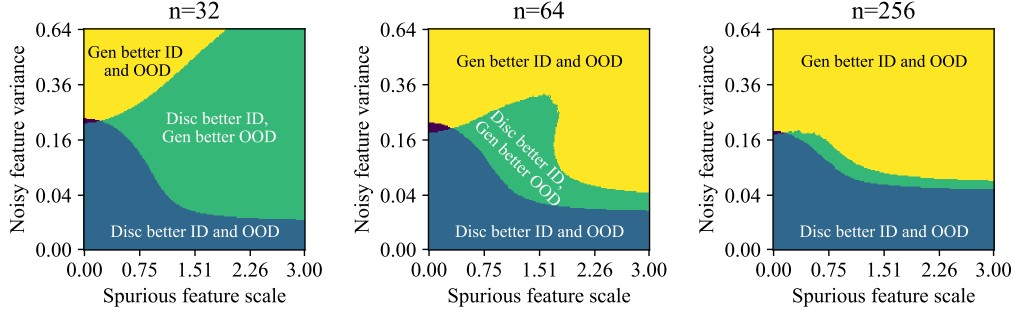

Figure 16: Each plot corresponds to a different number $n$ of training examples.

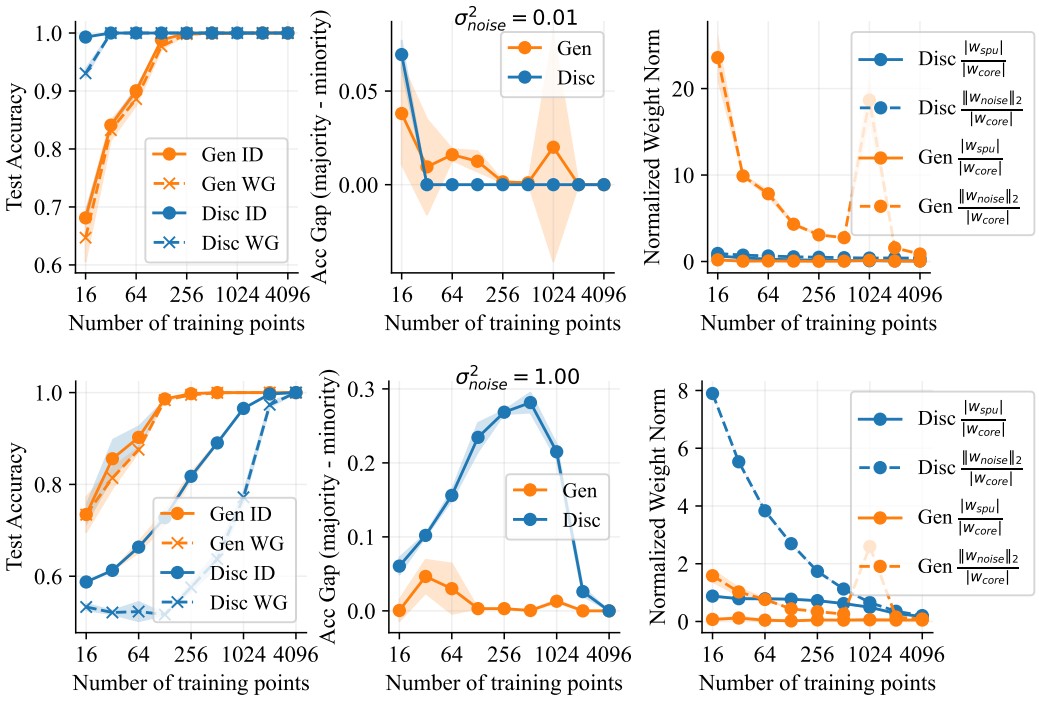

Figure 17: Effect of $\sigma_{noise}$ on the generalization of SVM vs LDA. Larger $\sigma_{noise}$ makes it easier for SVM to overfit, since it uses the high-norm noise features to increase its margin. Lower $\sigma_{noise}$ makes it harder to overfit, since the noise features are too small to significantly increase the margin.

## B    EXPERIMENTAL DETAILS

---
**Algorithm 1** `Generative Classifier`
---
1: **Input:** Training set $\mathcal{D} = \{(x_i, y_i)\}_{i=1}^N$
2: **Training model** $p_\theta(x|y)$**:**
3:     Minimize generative loss $\mathbb{E}_{(x,y)\sim\mathcal{D}}[-\log p_\theta(x|y)]$
4: **Classification of test input** $x$**:**
5: **for** class $\mathbf{y}_i \in \mathcal{Y}$ **do**
6:     Compute $p_\theta(x|y_i)$
7: **end for**
8: Return $\arg\max_{y_i} p_\theta(x|y_i)p(y_i)$

---

### B.1    IMAGE-BASED EXPERIMENTS

#### B.1.1    DIFFUSION-BASED GENERATIVE CLASSIFIER

We train diffusion models from scratch in a lower-dimensional latent space (Rombach et al., 2022). We use the default 395M parameter class-conditional UNet architecture and train it from scratch with AdamW (Loshchilov & Hutter, 2017) with a constant base learning rate of 1e-6 and no weight decay or dropout. We did not tune diffusion model hyperparameters and simply used the default settings for conditional image generation. Again, we emphasize: *we achieved SOTA accuracies under distribution shift, using the default hyperparameters from image generation.*

Each diffusion model requires about 3 A6000 days to train. For inference on Waterbirds, CelebA, and Camelyon, we sample 100 noises $\epsilon$ and use them with each of the two classes. For FMoW, we

use the adaptive strategy from Diffusion Classifier (Li et al., 2023) that uses 100 samples per class, then does an additional 400 samples for the top 5 remaining classes.

### B.1.2 DISCRIMINATIVE BASELINES

We use the official training codebase released by Koh et al. (2021) to train our discriminative baselines. For image-based benchmarks, we train 3 model scales (ResNet-50, ResNet-101, and ResNet-152) and sweep over 4 learning rates and 4 weight decay parameters. We use standard augmentations: normalization, random horizontal flip, and RandomResizedCrop.

### B.2 AUTOREGRESSIVE GENERATIVE CLASSIFIER

For training, we pad shorter sequences to a length of 512 and only compute loss for non-padded tokens. We use a Llama-style architecture (Touvron et al., 2023) and train 15M and 42M parameter models from scratch. We train for up to 200k iterations, which can take 2 A6000 days. We use a repository without default hyperparameters, so we sweep over learning rate, weight decay, and dropout based on their effect on the data log-likelihood. The resulting family of models is then shown in Figure 2.

### B.2.1 DISCRIMINATIVE BASELINES

For CivilComments, we use a randomly initialized encoder-only transformer with the same architecture as DistilBert (Sanh et al., 2019). We train for 100 epochs and sweep over dropout rate, learning rate, and weight decay.

