# OpenReview forum: "Generative Classifiers Avoid Shortcut Solutions"
_ICLR.cc/2025/Conference — ICLR 2025 Poster_

### Official Review · Reviewer_vNha · 2024-10-24

**Soundness:** 3
**Presentation:** 3
**Contribution:** 3
**Rating:** 8
**Confidence:** 4

**Summary:**

This paper revisits generative classifiers to address the problem of spurious features that affect discriminative classifiers trained with cross-entropy loss on out-of-distribution data. Leveraging class-conditional generative models, based on diffusion for image related problems and autoregressive models for text related problems, the authors show that generative classifiers outperform discriminative classifier both in-distribution and out-of-distribution on various benchmarks. The authors also propose some analysis on a toy task.

**Strengths:**

* While the concept of generative classifiers is not new, applying recent generative models to the challenging out-of-distribution generalization problem is timely and interesting. The paper is well written and has a strong overall message.

* The paper provides evaluation on several datasets covering both image and text-related problems. The improvement due to generative classifiers over discriminative baselines seems important in both out-of-distribution and even in-distribution scenarios.

* The authors provided intuitions why the generative classifiers performs better; in particular I found the analysis on the gradient norm and the toy evaluation well lead and insightful.

**Weaknesses:**

* Comparison to classifier with pretraining and data augmentation: although pretraining does not seem to improve the proposed model, it does have a big importance for standard classifiers; in particular, could the authors show (e.g. on one chosen image-based dataset) that finetuning a pretrained ResNet-50 classifiers (on imagenet) does not reduce significantly the gap with the considered approach ? Moreover, applying data augmentation is also a standard trick for image-based classification. Although its application to the generative classifiers is still unclear (as mentioned in the conclusion), was it included in the discriminative models reported in the paper ?
* Multi-class classification: the authors seem to have mostly included binary classification tasks; does the proposed model scale to multi-class classification problems ? If yes, including some evaluation on a multi-class problem could add some value to the experimental evaluation (e.g. DomainBed - https://arxiv.org/abs/2007.01434 - provides some multi-class image-based benchmarks where comparison can be made with several invariant based approaches).

**Questions:**

See above.

===Post rebuttal===

I thank the authors for the rebuttal; I recommend acceptance.

---

> ### Author Response · Authors · 2024-11-30
>
> We really appreciate your comments and suggestions! We answer your questions below:
>
> > Comparison to classifier with pretraining and data augmentation: although pretraining does not seem to improve the proposed model, it does have a big importance for standard classifiers
>
> To be clear, we didn’t find that “pretraining does not seem to improve the proposed model”; we believed it would be difficult to fairly match the strength of pretraining across all methods, and had decided to train all models from scratch instead. Based on your suggestion, we’ve now added new results to Appendix A.7 that show pretrained discriminative models do not achieve the same “effective robustness” (better OOD accuracy for any given ID performance) as even a generative classifier trained from scratch.
>
> > Was [data augmentation] included in the discriminative models reported in the paper?
>
> We train all image models in this paper (discriminative and generative) with the same set of standard augmentations: color normalization, random horizontal flip, and RandomResizedCrop. We’ve clarified this in our section on training details (Appendix B.1.2). We’ve also added results with stronger augmentations for discriminative models on several new benchmarks (see next point), and find that they do not close the “effective robustness” gap with generative models.
>
> > Does the proposed model scale to multi-class classification problems?
>
> Yes, generative classifiers do work on multi-class classification problems! Our current results on FMoW show that generative classifiers work on 62-way classification. To be thorough, we’ve now added results on the Living-17 and Entity-30 distribution shifts from the popular BREEDS benchmark [1]. These datasets have 17 and 30 classes, respectively. We find that generative classifiers again display “effective robustness” compared to discriminative models. We also use the results reported in the paper as strong discriminative baselines, which include stronger techniques such as adversarial training or stronger augmentations (RandErase, gaussian noise, or stylization). Our generative classifiers achieve better OOD accuracy than any of these interventions. Thank you for your suggestion to try additional multiclass datasets!
>
> Please let us know if you have any further questions!
>
> [1] Santurkar, S., Tsipras, D., & Madry, A. (2020). Breeds: Benchmarks for subpopulation shift. arXiv preprint arXiv:2008.04859.

---

> > ### Comment · Reviewer_vNha · 2024-12-01
> >
> > I thank the authors for the rebuttal; I increased my score.

---

### Official Review · Reviewer_5wra · 2024-11-01

**Soundness:** 3
**Presentation:** 3
**Contribution:** 4
**Rating:** 8
**Confidence:** 3

**Summary:**

The paper shows that generative classifiers always perform better in the realistic OOD setting, which can be implemented easily based on existing class-conditional generative models. They conduct simulations to show that generative classifiers can model both core and spurious, instead of mainly spurious ones. They also present the data properties that affect when generative classifiers outperform discriminative ones.

**Strengths:**

1. The paper is novel and well-written.
2. The paper shows that generative classifiers have advantages compared to discriminative classifiers, both in ID and OOD settings. I think this is a fundamental challenge to existing discriminative learning, which is significant.
3. The experiments are sufficient to validate the proposed hypothesis.

**Weaknesses:**

1. I think some recent papers about the superiority of generative classifiers should be mentioned in this paper, including [a,b,c].

[a] Diffusion Models are Certifiably Robust Classifiers, NeurIPS, 2024

[b] Robust Classification via a Single Diffusion Model, ICML, 2024

[c] Revisiting Discriminative vs. Generative Classifiers: Theory and Implications, ICML, 2023

**Questions:**

1. In line 322, why do you choose epoch 5? Can this affect the final results?
2. In line 341, the increase of gradient norms makes me confused. It should decrease during the training because the gradients will converge to 0.
3. In line 364, the paper claims that parameter count does not seem to be responsible for the performance of the generative classifier. Does this mean, when considering downstream discriminative tasks, we can not enjoy the scaling law of generative foundation models?

If you can address the weaknesses and questions sufficiently, I will improve the rating.

---

> ### Author Response · Authors · 2024-11-30
>
> We really appreciate your comments and suggestions! We answer your questions below:
>
> > I think some recent papers about the superiority of generative classifiers should be mentioned in this paper
>
> Thank you for the recommendations! The two papers on improving adversarial robustness with generative classifiers are definitely related; we do note that previous work has found that adversarial training has not improved robustness to distribution shift  [1,2,3], so it’s not expected based on these two papers that generative classifiers would be better at distribution shift. We’ve added citations to these papers and made a comment to readers about this, thanks!
>
> > In line 322, why do you choose epoch 5? Can this affect the final results?
>
> We previously chose epoch 5 in order to avoid any “warmup effect” at the very beginning of training. In practice, we double checked, and there is no qualitative difference in these plots if we use epoch 1 vs epoch 5.
>
> > In line 341, the increase of gradient norms makes me confused. It should decrease during the training because the gradients will converge to 0.
>
> We plot the average of the per-example gradient norms $\frac{1}{n} \sum_i ||\nabla_\theta \mathcal L (x_i, y_i)||$, not the norm of the average gradient $||\frac{1}{n} \sum_i \nabla_\theta \mathcal L (x_i, y_i)||$. For generative models, the latter quantity converges to 0, but the per-example gradient norms do not (since there is always a way to increase the likelihood of a single data point). This is good, because it means that each datapoint always provides useful learning signal throughout training, as opposed to the learning signal rapidly decaying to 0 (which happens for discriminative models).
>
> > In line 364, the paper claims that parameter count does not seem to be responsible for the performance of the generative classifier. Does this mean, when considering downstream discriminative tasks, we can not enjoy the scaling law of generative foundation models?
>
> This is a really good question! To summarize Section 5.3, we found that increasing *discriminative model size* does not improve performance. This limitation for discriminative models has also been previously observed and theoretically characterized [4]. Based on your suggestion, we’ve now added scaling results for generative classifiers in Appendix A.4 and Fig. 13, and we observe that generative classifiers can benefit a lot from scaling up model size. We really appreciate your comment, and think that this finding that “generative classifiers benefit more from scaling than discriminative classifiers do” is quite interesting!
>
> [1] Taori, R., Dave, A., Shankar, V., Carlini, N., Recht, B., & Schmidt, L. (2020). Measuring robustness to natural distribution shifts in image classification. Advances in Neural Information Processing Systems, 33, 18583-18599.
>
> [2] Santurkar, S., Tsipras, D., & Madry, A. (2020). Breeds: Benchmarks for subpopulation shift. arXiv preprint arXiv:2008.04859.
>
> [3] Miller, J. P., Taori, R., Raghunathan, A., Sagawa, S., Koh, P. W., Shankar, V., ... & Schmidt, L. (2021, July). Accuracy on the line: on the strong correlation between out-of-distribution and in-distribution generalization. In International conference on machine learning (pp. 7721-7735). PMLR.
>
> [4] Sagawa, S., Raghunathan, A., Koh, P. W., & Liang, P. (2020, November). An investigation of why overparameterization exacerbates spurious correlations. In International Conference on Machine Learning (pp. 8346-8356). PMLR.

---

> > ### Comment · Reviewer_5wra · 2024-12-02
> >
> > The authors have addressed most of my concerns, so I increase the rating from 6 to 8.

---

### Official Review · Reviewer_RwsJ · 2024-11-01

**Soundness:** 2
**Presentation:** 3
**Contribution:** 1
**Rating:** 5
**Confidence:** 4

**Summary:**

The paper explored the ability of generative classifiers to handle domain shifts. They use diffusion models and AR models for images and text data and show significant improvement in OOD classification with their classifier compared to other baselines.

**Strengths:**

- The paper handles the important topic of OOD generalization in machine learning
- The method shows good performance on several benchmarks
- The paper is well written

**Weaknesses:**

- The main issue is limited novelty. Using generative models for classification isn't a new idea, and previous works already explored their robustness to OOD generalization and to adversarial attacks (and also the use of diffusion models as classifiers. For example Zimmermann et al "Score-Based Generative Classifiers", Grathwohl et al "YOUR CLASSIFIER IS SECRETLY AN ENERGY BASED
MODEL AND YOU SHOULD TREAT IT LIKE ONE", Chen et al "Robust Classification via a Single Diffusion Model" and Fetaya et al "Understanding the Limitations of Conditional Generative Models". The paper should also add them as related works.

- While I like the use of simple toy examples, I don't think there was any insight or useful observation from the whole section. For example, the observation that "As σ increases, the core feature becomes less reliable, and the generative classifier uses the spurious and noise features more" is quite expected and doesn't give any added value. This section also is only relevant to generalization so it mirrors the results in Ng&Jordan 2001.

- Also regarding the Illustrative setting, the feature you name $x_{spu}$ is not a spurious feature as it has a direct causal relationship by the label (as you designed it so). It just isn't as useful for discrimination as the "core" feature.

- Regarding model size in the experiments, table 2 is also on CivilComments so that only covers one dataset. Moreover, on this dataset you performed worse than the discriminative models on the ID while you performed better than the other models on 3/4 of image datasets. This shows that the CivilComments behavior might not transfer. Also, parameter count isn't an exact comparison.

- In two of the datasets, waterbirds and FMoW you can see in Fig. 2 that the ID and OOD seem to be very highly correlated when looking over all models. Looking at these plots, especially FMoW, it is not clear at all that the better OOD performance isn't just because your base model had better ID performance and not because of some innate property of generative models.


Small remarks:
- L28 you write "Ever since AlexNet (Krizhevsky et al., 2012), classification has mainly been tackled with discriminative methods" but this isn't accurate as discriminative models have been the standard approach even before AlexNet, just mainly SVMs/decision trees/boosting etc.

**Questions:**

NA

---

> ### Author Response · Authors · 2024-11-30
>
> Thank you for your feedback! We’ve incorporated your suggestions into our paper, and answer your questions below:
>
> > The main issue is limited novelty. Using generative models for classification isn't a new idea, and previous works already explored their robustness to OOD generalization and to adversarial attacks (and also the use of diffusion models as classifiers. For example Zimmermann et al "Score-Based Generative Classifiers", Grathwohl et al "YOUR CLASSIFIER IS SECRETLY AN ENERGY BASED MODEL AND YOU SHOULD TREAT IT LIKE ONE", Chen et al "Robust Classification via a Single Diffusion Model" and Fetaya et al "Understanding the Limitations of Conditional Generative Models". The paper should also add them as related works.
>
> Thanks for the pointers to these papers — we’ve added them to our related works section along with a short discussion. However, we’d like to point out the novel contributions in our paper:
> - We show that generative classifiers are particularly *robust to distribution shift*. This has not been found in previous work. Zimmerman et al. [1] found that their generative classifier performed *worse* on distribution shift than discriminative models. Grathwohl et al. [2] and Chen et al. [3] only found improved adversarial robustness for generative classifiers, but had no results on distribution shifts. Fetaya et al. [4]’s results go in the other direction and indicate that generative classifiers may not be more adversarially robust in the first place. Finally, it’s been well-documented in the distribution shift literature that adversarial robustness has not translated to any robustness to distribution shift [5,6,7]. For these reasons, it was not clear from previous work that generative classifiers would be more robust to distribution shift, and this is our most important contribution.
> - We perform extensive experiments to understand why generative classifiers have better out-of-distribution generalization than discriminative classifiers (Section 5.3, Section 6, Appendix A.1, A.2, A.4, A.6).
> - We provide better understanding of the inductive biases of generative classifiers in a simplified Gaussian setting, which has not previously been done for generative classifiers. This understanding helps us understand when generative classifiers are better for distribution shift problems (not always!). Previous work has focused [8] has extensively analyzed this for discriminative models, but none have analyzed OOD generalization for generative classifiers.
>
> > the observation that "As σ increases, the core feature becomes less reliable, and the generative classifier uses the spurious and noise features more" is quite expected and doesn't give any added value. This section also is only relevant to generalization so it mirrors the results in Ng&Jordan 2001.
>
> Our observation that generative classifiers prefer reliable features is intuitive, but it has an important implication for OOD generalization. Consistent features are the ones that distributionally robust classifiers should use, and generative classifiers have the inductive bias to prefer using them! In contrast, discriminative classifiers have no such preference and thus suffer more from distribution shift. Furthermore, Ng and Jordan, which we cite, only analyze the convergence rate of the ID accuracy for Naive Bayes. It provides no insights into OOD generalization or the properties of the data that affect robustness.
>
> > the feature you name $x_{spu}$ is not a spurious feature as it has a direct causal relationship by the label (as you designed it so). It just isn't as useful for discrimination as the "core" feature.
>
> In our illustrative setting, we call $x_{spu}$ a spurious feature because it cannot, by itself, be used to classify the label 100% accurately. This is intended to model features like background color, which are correlated with the label at some high rate, but do not always hold (especially under distribution shift). Previous distribution shift literature with a similar Gaussian data setup has consistently called this a spurious feature [9,8,10,11]. We’re happy to add more text to clarify this in the paper!

---

> ### Author Response · Authors · 2024-11-30
>
> > Regarding model size in the experiments, table 2 is also on CivilComments so that only covers one dataset. Moreover, on this dataset you performed worse than the discriminative models on the ID while you performed better than the other models on 3/4 of image datasets. This shows that the CivilComments behavior might not transfer. Also, parameter count isn't an exact comparison.
>
> Figure 4 in our paper shows that scaling up the discriminative model does not improve performance at all on Waterbirds. Based on your suggestion, we’ve also added Fig. 12 to the Appendix, which shows the same conclusions on another 2 of our datasets. Finally, the CivilComments results in Table 2 should be a fair comparison, since the discriminative and generative models there have the same parameter count and architecture. Overall, we are quite confident that discriminative model size is not a confounder in our experiments.
>
> > In two of the datasets, waterbirds and FMoW you can see in Fig. 2 that the ID and OOD seem to be very highly correlated when looking over all models. Looking at these plots, especially FMoW, it is not clear at all that the better OOD performance isn't just because your base model had better ID performance and not because of some innate property of generative models.
>
> We’re happy to clarify this! Previous work has found that ID and OOD accuracy are strongly correlated across almost all distribution shift benchmarks [5,7]. This phenomenon is called “accuracy on the line” — models that do better ID will do predictably better OOD. The most interesting previous finding is that all interventions (changing model architecture or size, augmentations, adversarial training) that affect OOD accuracy only do so by affecting ID accuracy and moving the model along the line. Only adding more diverse training data (which reduces or eliminates the distribution shift) has been found to move models vertically *above* the line [5,12]. This is called “effective robustness,” and it indicates that these models are truly more innately robust to distribution shift than the baseline models that lie on the line.
>
> Our finding is that generative classifiers are the first approach to show “effective robustness,” and we observe it on most of our datasets (CelebA, CivilComments, and Living-17 and Entity-30 in our new results in Fig. 14). Generative classifiers don’t always have effective robustness (as you pointed out on Waterbirds or FMoW and as we wrote on L302). This agrees with the predictions from our “generalization phase diagrams” — generative classifiers will sometimes just be better both ID and OOD, based on the properties of the data. Overall, we think it’s exciting that generative classifiers are the first method to show effective robustness without using extra training data.
>
> > L28 you write "Ever since AlexNet (Krizhevsky et al., 2012), classification has mainly been tackled with discriminative methods" but this isn't accurate as discriminative models have been the standard approach even before AlexNet, just mainly SVMs/decision trees/boosting etc.
>
> Thanks for pointing this out — we’ve modified this to be more clear that we’re referring to classification with neural networks.
>
> Please let us know if you have any further questions!

---

> > ### Author Response · Authors · 2024-11-30
> >
> > References:
> >
> > [1] Zimmermann, R. S., Schott, L., Song, Y., Dunn, B. A., & Klindt, D. A. (2021). Score-based generative classifiers. arXiv preprint arXiv:2110.00473.
> >
> > [2] Grathwohl, W., Wang, K. C., Jacobsen, J. H., Duvenaud, D., Norouzi, M., & Swersky, K. (2019). Your classifier is secretly an energy based model and you should treat it like one. arXiv preprint arXiv:1912.03263.
> >
> > [3] Chen, H., Dong, Y., Wang, Z., Yang, X., Duan, C., Su, H., & Zhu, J. (2023). Robust classification via a single diffusion model. arXiv preprint arXiv:2305.15241.
> >
> > [4] Fetaya, E., Jacobsen, J. H., Grathwohl, W., & Zemel, R. (2019). Understanding the limitations of conditional generative models. arXiv preprint arXiv:1906.01171.
> >
> > [5] Taori, R., Dave, A., Shankar, V., Carlini, N., Recht, B., & Schmidt, L. (2020). Measuring robustness to natural distribution shifts in image classification. Advances in Neural Information Processing Systems, 33, 18583-18599.
> >
> > [6] Santurkar, S., Tsipras, D., & Madry, A. (2020). Breeds: Benchmarks for subpopulation shift. arXiv preprint arXiv:2008.04859.
> >
> > [7] Miller, J. P., Taori, R., Raghunathan, A., Sagawa, S., Koh, P. W., Shankar, V., ... & Schmidt, L. (2021, July). Accuracy on the line: on the strong correlation between out-of-distribution and in-distribution generalization. In International conference on machine learning (pp. 7721-7735). PMLR.
> >
> > [8] Nagarajan, V., Andreassen, A., & Neyshabur, B. (2020). Understanding the failure modes of out-of-distribution generalization. arXiv preprint arXiv:2010.15775.
> >
> > [9] Sagawa, S., Raghunathan, A., Koh, P. W., & Liang, P. (2020, November). An investigation of why overparameterization exacerbates spurious correlations. In International Conference on Machine Learning (pp. 8346-8356). PMLR.
> >
> > [10] Idrissi, B. Y., Arjovsky, M., Pezeshki, M., & Lopez-Paz, D. (2022, June). Simple data balancing achieves competitive worst-group-accuracy. In Conference on Causal Learning and Reasoning (pp. 336-351). PMLR.
> >
> > [11] Setlur, A., Dennis, D., Eysenbach, B., Raghunathan, A., Finn, C., Smith, V., & Levine, S. (2023). Bitrate-constrained DRO: Beyond worst case robustness to unknown group shifts. arXiv preprint arXiv:2302.02931.
> >
> > [12] Fang, A., Ilharco, G., Wortsman, M., Wan, Y., Shankar, V., Dave, A., & Schmidt, L. (2022, June). Data determines distributional robustness in contrastive language image pre-training (clip). In International Conference on Machine Learning (pp. 6216-6234). PMLR.

---

> ### Author Response · Authors · 2024-12-02
>
> We hope we have addressed all your questions. If there are any other concerns remaining that prevent you from accepting the paper, please let us know!

---

> > ### Comment · Reviewer_RwsJ · 2024-12-02
> > **Response to authors**
> >
> > - I accept some of their claim about novelty, however, the novelty is still somewhat limited. The authors should discuss why there are some contradictory results compared to Zimemrman et al.
> >
> > - I disagree with the importance of the toy example, but as importance is a subjective measure there isn't much to discuss.
> >
> > - Using Wikipedia, "In statistics, a spurious relationship or spurious correlation[1][2] is a mathematical relationship in which two or more events or variables are associated but not causally related, due to either coincidence or the presence of a certain third, unseen factor (referred to as a "common response variable", "confounding factor", or "lurking variable").". The example you give in the illustrative example isn't a spurious relationship as there is a direct causal relationship between the label and $x_{spu}$ (even if it is noisy). I would highly recommend changing the name to avoid misunderstandings
> >
> > - Your discussion about effective robustness is interesting, it would be best if the ID results and the discussion where added to the paper.
> >
> > I will raise my score given the rebuttal.

---

### Official Review · Reviewer_itDA · 2024-11-04

**Soundness:** 3
**Presentation:** 3
**Contribution:** 2
**Rating:** 6
**Confidence:** 3

**Summary:**

The paper explores the performance of generative classifiers as compared to discriminative classifiers under distribution shift. It first documents the performance of both types of models on 5 standard benchmarks for distribution shifts concluding that generative classifiers indeed perform better under the distribution shift but surprisingly also often in in-distribution classification. It argues that this advantage stems from the generative classifiers having to learn all instead of just the short-cut features for max-margin classification induced by the cross-entropy loss. It further explores the behavior of the two classifier types on a synthetic toy problem controlling for the ratios and variance of spurious, core and noise features and provides useful insights into the effects of these data properties through outlining "generatlization phase diagrams". This investigation leads to no free lunch conclusion stating that the advantage of generative vs discriminative classifier is a function of the data properties which is in line with the results of the benchmark experimentation.

**Strengths:**

The empirical investigation in the paper corroborates some rather well-accepted intuitions of potential benefits of generative classifiers in distribution-shift scenarios and thus provides useful support for future work in this direction. The systematic toy-problem exploration helps to understand the trade-offs and uses some tools that could be adapted for exploration of more complex scenarios in the future. Some further useful results are presented in the appendix (perhaps worth at least referencing in the main text?). The paper is well balanced in its conclusions, contributing to general understanding of the investigated phenomenons. The paper is well written and easy to follow.

**Weaknesses:**

It is unclear to me, how the conclusions documented on the rather simplistic and well controlled toy-problem could be translated to more complex practical problems under realistic scenarios. This is, however, acknowledged by the authors themselves.

**Questions:**

1. The superior ID performance of generative models in Table 1 suggests to me that perhaps the discriminative models are too weak. For example, the original Sagawa paper reported the ID accuracy of discriminative classifiers for the Waterbird dataset between 94-97% (depending on regularization) and close to 95% for the CelebA. If this were true, to comparison would be unfair and not very useful. This feeling of "unfair / skewed comparison" is further reinforced by the comment in page 7 - discriminative classifier for CivilComments is larger and it indeed performs better on ID. Can you comment and clarify, how you ensure the fairness of the comparisons? If the comparison is indeed not fair, what does this mean for the presented results and the conclusions we can take from it?
2. In page 5 you speak about "class-balanced accuracy". I am not sure how this is defined. Can you please clarify?
3. Could  not the evolution of the gradient norm in Figure 3 simply suggest that the discriminative classifier learns faster?
4. Section 6.3 Figure 6 - you fix $\sigma^2_{noise} = 0.36$. Why particularly this value? Is this somehow representative of the behavior? Would the graphs look similarly at different noise levels?
5. Section 6.3 Figure 6 uses the feature scale B=1? Or some other value?

---

> ### Author Response · Authors · 2024-11-30
>
> Thank you for your feedback! We’ve incorporated your suggestions into our paper, and answer your questions below:
>
> > It is unclear to me how the conclusions documented on the rather simplistic and well controlled toy-problem could be translated to more complex practical problems under realistic scenarios.
>
> You’re right that the simple setting is quite different from the reality of complex distribution shifts. However, we do still think that there are interesting takeaways from our illustrative setting. First, the biggest hallmark of a good toy setting is that it makes useful predictions for the real world — and we do see that predictions here are borne out by neural nets on real benchmarks. In particular, the “generalization phase diagrams” in this illustrative setting show that different data properties can induce (a) discriminative models are better ID and OOD, (b) discriminative models are better ID but generative models are better OOD, or (c) generative models are better both ID and OOD. However, it almost never predicts a scenario where discriminative models do better OOD, but generative models do better ID. And indeed, this matches our neural network results in Section 5. Second, our toy setting also makes it easy to control various data properties (strength of spurious correlation, ease of overfitting, consistency of core feature) and understand how much a model relies on each kind of feature. Finally, it could be a useful setting for theorists in the future to derive generalization bounds or more principled algorithms with more favorable inductive biases.
>
> > Can you comment and clarify how you ensure the fairness of the comparisons?
>
> For a completely fair comparison, we trained all discriminative and generative models from scratch. This is because using an ImageNet-pretrained checkpoint, as is done in the original Sagawa paper, completely removes the need for feature learning on many of these tasks. For example, Waterbirds becomes trivial, since pretraining on ImageNet requires the model to differentiate between dozens of bird species, in a setting where the background is no longer a strong spurious correlation.
>
> To ensure that the discriminative baseline is as strong as possible, we use the official code from Koh, Sagawa et al. [1] and we sweep over 3 model sizes (ResNet-50, ResNet-101, and ResNet-152), 4 learning rates, and 4 weight decays. For this reason, we are confident that this represents a fair comparison. We have added more information to Appendix B.1 to clarify this.
>
> > In page 5 you speak about "class-balanced accuracy". I am not sure how this is defined.
>
> Class-balanced accuracy refers to computing the accuracy for each class independently, and then uniformly averaging those per-class accuracies into a single number. This reduces the effect of class imbalance in the validation data, which has previously been found to exacerbate models’ susceptibility to spurious correlations.
>
> > Could not the evolution of the gradient norm in Figure 3 simply suggest that the discriminative classifier learns faster?
>
> In some sense, yes, and this is the problem! The discriminative classifier quickly “achieves its objective” of minimizing the discriminative loss by learning a *shortcut solution* on the majority examples where the spurious correlation holds. The discriminative objective can be easily cheated because it does not require models to learn the core features that will be consistent across distribution shifts. In contrast, the generative classifier has a harder objective that takes longer to learn, but this does encourage the model to eventually learn and use the core features.
>
> > Section 6.3 Figure 6 - you fix $\sigma_{noise}^2=0.36$. Why particularly this value? Is this somehow representative of the behavior? Would the graphs look similarly at different noise levels?
>
> Good question! We used $\sigma_{noise}^2 = 0.36$ to illustrate a setting where SVM and LDA classifiers achieve roughly the same in-distribution accuracy, yet LDA achieves much better out-of-distribution accuracy. We thought this was particularly interesting because it corroborates our “effective robustness” results on CelebA and CivilComments in Figure 2, as well as new results on 2 additional datasets that we’ve added in Fig. 14.
>
> As we previously showed in our “generalization phase diagrams,” increasing $\sigma_{noise}$ hurts SVM’s performance, and decreasing $\sigma_{noise}$ improves SVM’s performance. Based on your suggestion, we’ve added equivalents of Fig. 6 to the Appendix (Fig. 11) to highlight these two cases.
>
> > Section 6.3 Figure 6 uses the feature scale B=1? Or some other value?
>
> Yes, we use feature scale B=1 for the Figures in this section.

---

> > ### Author Response · Authors · 2024-11-30
> >
> > We also think you might also be interested in some new results that we’ve added:
> > - *Generative classifiers benefit from parameter scaling* (Fig. 13). Scaling the number of model parameters has drastically improved generative modeling performance, in settings like language modeling and image/video generation. We demonstrate that generative *classifiers* also benefit from these scaling trends.
> > - *Effective robustness on new datasets* (Fig. 14). We additionally train generative classifiers on the Living-17 and Entity-30 distribution shifts from the popular BREEDS benchmark [2]. Generative classifiers again display “effective robustness” (better OOD accuracy for any given ID performance) compared to discriminative models. We also use the results reported in [2] as discriminative baselines, which includes stronger techniques such as adversarial training or stronger augmentations. Generative classifiers achieve better OOD accuracy than any of these interventions.
> >
> > Please let us know if you have any remaining questions!
> >
> > [1] Koh, P. W., Sagawa, S., Marklund, H., Xie, S. M., Zhang, M., Balsubramani, A., ... & Liang, P. (2021, July). Wilds: A benchmark of in-the-wild distribution shifts. In International conference on machine learning (pp. 5637-5664). PMLR.
> >
> > [2] Santurkar, S., Tsipras, D., & Madry, A. (2020). Breeds: Benchmarks for subpopulation shift. arXiv preprint arXiv:2008.04859.

---

> ### Author Response · Authors · 2024-12-03
>
> We hope we have addressed all your questions. If there are any other concerns remaining, please let us know!

---

> > ### Comment · Reviewer_itDA · 2024-12-03
> >
> > Thank you for all your clarification. In case of acceptance, please include the clarifications from your responses into the manuscript (or appendix).
> > I maintain my score.

---

### Meta-Review · Area_Chair_jzGS · 2024-12-16

**Metareview:**

The paper studies the performance of generative classifiers for their generalization under distribution shift compared to standard discriminative classifiers. I believe this is an important problem to study especially now as the field is moving more and more towards repurposing generative models for downstream discriminative tasks. The authors provide strong empirical results and break the problem down with small scale but illustrative examples.

Reviewers overall thought the paper was well written and thought the experimental results were of high quality although they had concerns about the theoretical claims generalizing to real-world settings and other small concerns about the experimental setup. Throughout the rebuttal process most of the concerns were alleviated.

I believe this is a quality work and should be accepted.

**Additional Comments On Reviewer Discussion:**

Reviewers raised concerns about the novelty of the work compared to prior works on the topic. The authors added additional citations and comparison to prior work. They added additional experiments and were attentive to reviewers questions. This caused most reviewers to increase their scores and there were general agreement about the quality of the work.

---

### Decision · Program_Chairs · 2025-01-22

Accept (Poster)